

# An optimization algorithm for maximum quasi-clique problem based on information feedback model

Shuhong Liu[1], Jincheng Zhou[2,3], Dan Wang[3], Zaijun Zhang[3] and Mingjie Lei[4]

[1] State Key Laboratory of Public Big Data, College of Computer Science and Technology, Guizhou University, Guiyang, Guizhou, China
[2] Key Laboratory of Complex Systems and Intelligent Optimization of Guizhou Province, Duyun, Guizhou, China
[3] School of Computer and Information, Qiannan Normal University for Nationalities, Duyun, Guizhou, China
[4] School of Earth Science and Surveying Engineering, China University of Mining and Technology (Beijing), Beijing, China

## ABSTRACT

The maximum clique problem in graph theory is a well-known challenge that involves identifying the complete subgraph with the highest number of nodes in a given graph, which is a problem that is hard for nondeterministic polynomial time (NP-hard problem). While finding the exact application of the maximum clique problem in the real world is difficult, the relaxed clique model quasi-clique has emerged and is widely applied in fields such as bioinformatics and social network analysis. This study focuses on the maximum quasi-clique problem and introduces two algorithms, NF1 and NR1. These algorithms make use of previous iteration information through an information feedback model, calculate the information feedback score using fitness weighting, and update individuals in the current iteration based on the benchmark algorithm and selected previous individuals. The experimental results from a significant number of composite and real-world graphs indicate that both algorithms outperform the original benchmark algorithm in dense instances, while also achieving comparable results in sparse instances.

## INTRODUCTION

Given a graph, a clique is a subset of vertices where each pair of vertices is adjacent, which is a fundamental concept in graph theory. The Maximum Clique Problem (MCP) is a well-known problem in this field (*Akhtanov et al., 2022*). However, in practical applications, the strict definition of clique has significant limitations because it requires direct adjacency between all pairs of vertices. To address this problem, the maximum $\gamma$-quasi-clique problem (MQCP) extends the MCP and finds applications in various fields such as complex network analysis (*Tenekeci & Isik, 2020*), clustering (*Chen et al., 2019*), and bioinformatics. For instance, when searching for groups on social networks (*Conde-Cespedes, Ngonmang & Viennet, 2018*), members may not be directly connected but can still form a closely

Corresponding author
Jincheng Zhou, zjc81@sgmtu.edu.cn

connected social circle as a whole. Using the maximum clique concept alone might overlook members with indirect connections in social network.

By utilizing the concept of relaxed cliques, specifically maximum $\gamma$-quasi-cliques, we can uncover these absent member connections. Real-world social networks exhibit intricate interpersonal relationships, and the maximum $\gamma$-quasi-clique model corresponds more effectively to the connectivity structures observed in social networks.

Given a graph $G = (V, E)$ and a fixed constant $\gamma \in (0, 1]$, $V$ and $E$ are the vertex and edge sets of graph $G$, respectively. $d_s(v)$ representing the degree of vertex $v$ in solution set $S$, and $N_G(v)$ are the adjacent vertex sets of vertex $v$ in graph $G$. A $\gamma$-quasi-clique $G[S]$ is a graph induced by a subset $V$ of vertices, such that $dens(G[S]) \geq \gamma$, where $\gamma \in (0, 1]$ is a fixed constant, $dens(G) = |E|/(|V| * (|V| - 1)/2)$. The goal of MQCP is to identify the largest $\gamma$-quasi-clique $S_{best}$ in the graph. MCP is a special case of MQCP. For each fixed constant $\gamma \in (0, 1]$, MQCP is an NP-hard problem.

## RELATED WORK

There are two main types of MQCP algorithms: exact algorithms and heuristic algorithms. In recent years, several exact algorithms have been introduced to address the MQCP (*Djeddi, Haddadene & Belacel, 2019*). However, these algorithms are limited to handling small or medium-sized problems due to their high time complexities. Conversely, heuristic algorithms like evolutionary algorithms can produce high-quality solutions for large-scale MQCPs within a reasonable time frame (*Pinto et al., 2018a*; *Pinto et al., 2018b*), although they do not guarantee optimality. To strike a balance between solution quality and time complexity, some hybrid algorithms combine both exact and heuristic elements (*Zhou, Liu & Gao, 2023*).

*Pattillo et al. (2013)* initially outlined key characteristics of the maximum $\gamma$-quasi-clique problem, demonstrating NP-completeness for fixed decision versions with $0 < \gamma < 1$, defining quasi-hereditary properties, and establishing an upper bound for analyzing maximum $\gamma$-quasi-clique size. They also presented a mixed integer programming (MIP) formulation and shared preliminary numerical results obtained through advanced solvers to identify exact solutions. Subsequently, *Pastukhov et al. (2018)* introduced a branch and bound algorithm for exact solutions to the maximum degree-based quasi-clique problem, alongside a degree-based decomposition heuristic. More recently, *Ribeiro & Riveaux (2019)* devised an exact enumeration algorithm that utilized the quasi-hereditary property, supporting backtracking search tree strategies. Additionally, they introduced a novel upper bound that consistently outperforms previous bounds, leading to a significant reduction in search space and enumerated subgraphs.

*Miao & Balasundaram (2020)* defined the maximum $\gamma$-quasi-clique problem as an optimization problem with a linear objective and a single quadratic constraint in binary variables. They explored the Lagrangian dual of this formulation and introduced an upper bounding technique using ellipsoid geometry to enhance the dual bound. Experimental results on standard benchmarks demonstrated that the upper bound surpasses the bounds obtained through mixed integer programming (MIP). One practical application of the

maximum clique problem is in identifying hidden quasi-cliques. *Abdulsalaam & Ali (2021)* focused on recovering planted quasi-cliques and introduced a convex nuclear norm minimization (NNM) formulation. Through numerical experiments utilizing existing mixed-integer programming formulations, the study found that the convex formulation outperforms the mixed-integer programming approach when $\gamma$ exceeds a specific threshold.

*Yu & Long (2023)* proposed a new branch-and-bound algorithm FastQC and its improved version DCFastQC, which significantly improved the computational efficiency of the maximum quasi-clique enumeration problem by combining improved pruning technology and new branching methods. It solves the efficiency bottleneck of existing exact algorithms in dealing with this NP-hard problem, and verifies the significant performance improvement of the new method on real data sets through a large number of experiments.

*Marinelli, Pizzuti & Rossi (2021)* proposed a new MIP reformulation $[D_\gamma]$ for the $\gamma$-quasi-clique problem, derived from decomposing star inequalities. This reformulation has shown similar performance to the tightest formulation $[C_\gamma]$ in existing literature. Experiments also indicate that the surrogate relaxation $[D_\gamma^S]$ produces comparable bounds. The task of cohesive subgraph mining in static graphs has been extensively studied in network analysis over the years. *Lin et al. (2021)* tackled this challenge by utilizing traditional quasi-cliques and introducing a new model called maximal $\rho$-stable $(\delta, \gamma)$-quasi-clique to evaluate subgraph cohesion and stability. They developed a novel temporal graph reduction algorithm to preserve all maximal $\rho$-stable $(\delta, \gamma)$-quasi-cliques while simplifying the temporal graph significantly. Furthermore, they introduced an efficient branch-and-bound enumeration algorithm, *BB &SCM*, for the simplified temporal graph.

Despite the optimality guarantees of exact algorithms, their feasibility is limited for large-scale quasi-clique instances. On the other hand, there are many fast heuristic algorithms that efficiently handle such instances, providing good approximate solutions within reasonable time frames. For example, *Khosraviani & Sharifi (2011)* introduced a distributed $\gamma$-quasi-clique extraction algorithm using the MapReduce model, demonstrating excellent scalability. The optimization of alternative density functions is crucial for finding dense subgraphs. One widely adopted function is average degree maximization, leading to the concept of the densest subgraph. *Tsourakakis et al. (2013)* defined a novel density function that surpasses the densest subgraph in generating higher-quality subgraphs. They also developed an additive approximation algorithm and a local search heuristic algorithm to optimize this novel density function.

*Mahdavi Pajouh, Miao & Balasundaram (2014)* were the first to develop a combined branch-and-bound algorithm for solving the maximum $\gamma$-quasi-clique problem, utilizing an upper bound technique. Their findings highlight the algorithm's competitive performance, especially in sparse random graph instances and DIMACS clique benchmark instances. *Lee & Lakshmanan (2016)* introduced the core tree concept, which organizes dense subgraphs recursively, reduces the search space, and assists in finding solutions through multiple tree traversals. They proposed three improved operations—add, delete, and exchange—to improve solutions, along with two iterative maximization algorithms, DIM and SUM, which approach QMQ deterministically and stochastically, respectively.

*Pinto et al. (2018a)* and *Pinto et al. (2018b)* proposed two variants of a biased random key genetic algorithm for solving the maximum quasi-clique problem and evaluated the corresponding algorithm variables. They enhanced an optimized iterative greedy algorithm by constructing a heuristic decoder to produce superior numerical results. The results of their calculations demonstrate that this novel method improves upon existing heuristic algorithms. In their subsequent research, *Pinto et al. (2021)* introduced a new approach to addressing the maximal cardinality quasi-clique problem. Their LSQClique algorithm, which combines a biased random-key genetic algorithm (BRKGA) with an exact local search strategy, presented an innovative solution. Alongside this algorithm, they introduced DECODER-LSQClique, a comprehensive method for solving the quasi-clique problem.

In a related study, *Zhou, Benlic & Wu (2020)* proposed the opposition-based memetic algorithm (OBMA), which utilizes a backbone-based crossover operator to create new offspring and incorporates a constrained neighborhood taboo search for local optimization. By integrating opposition-based learning (OBL), the algorithm enhances the search capabilities of traditional memetic algorithms, providing a robust solution. Additionally, *Peng et al. (2021)* introduced the hybrid artificial bee colony algorithm (HABC) to tackle the MQC problem. This novel approach integrates various specialized strategies into the artificial bee colony framework. The algorithm starts with an opposition-based initialization phase and then cycles through employed bees, onlooker bees, and scout bees phases iteratively to effectively conduct the search.

*Khalil et al. (2022)* introduced a parallel solution for extracting maximal quasi-cliques by utilizing the distributed graphics mining framework G-thinker. G-thinker is specifically tailored for single-machine multi-core environments to improve accessibility for regular end-users. In order to simplify the development of parallel applications, the researchers also created a general framework called T-thinker, focusing on a divide-and-conquer strategy. The study also addresses the difficulty of mining large quasi-cliques directly from the dense regions of a graph. They identified and effectively resolved the issue of repeated searches associated with the nearest method by implementing a well-designed concurrent data structure called a *trie*.

*Sanei-Mehri et al. (2021)* focused on enumerating top-degree-based quasi-cliques and highlighted the NP-hard nature of determining the largest quasi-clique. Their method involves identifying kernels of dense and large subgraphs, expanding subgraphs around these kernels to achieve the desired density. They proposed the kernelQC algorithm as a heuristic approach for enumerating the $k$-largest quasi-clique in a graph. In a related study, *Payne et al. (2021)* introduced the Automatic Quasi-Clique Merger (AQCM) algorithm, derived from the QCM (quasi-clique merger). AQCM performs hierarchical clustering in datasets, using a relevant similarity measure to assess the similarity between data sets.

Recently, *Chen et al. (2021)* proposed NuQClq, a local search algorithm that incorporates two innovative strategies. The first strategy addresses tie-breaking in the primary scoring function by introducing a new secondary scoring function. The second strategy overcomes local search stagnation by introducing a novel configuration checking strategy named

*BoundedCC*. Experimental results demonstrate the algorithm's superior performance, surpassing all existing methods.

In addition, there have been many studies on the quasi-clique problem in recent years. *Ali et al. (2023)* proposed a new hierarchical graph pooling method Quasi-Clique Pool, which is based on the concept of quasi-clique to extract dense incomplete subgraphs in the graph, and introduces a soft peeling strategy to preserve the topological structure relationship between nodes. Experimental results show that combining Quasi-Clique Pool with the existing graph neural network architecture improves the accuracy by an average of 2% on six graph classification benchmarks, which is better than other existing pooling methods. *Konar & Sidiropoulos (2024)* studied the optimal quasi-clique problem (OQC) and revealed that the density of the OQC solution obtained by continuous changes in parameter values is equal to that of the classic densest-$k$-subgraph problem. *Santos et al. (2024)* conducted a study on the multi-objective quasi-clique problem (MOQC), which aims to maximize both the number of vertices and the edge density of the quasi-clique. This problem is an extension of the single-objective quasi-clique problems, such as the maximum quasi-clique problem and the dense-$k$-subgraph problem. They discussed the relationship between the MOQC problem and the single-objective quasi-clique problem, and introduced some theoretical properties. Additionally, based on the $\varepsilon$-constraint method algorithm, a three-stage strategy method is proposed to solve the MOQC problem.

Elephant Herding Optimization (EHO) is a meta-heuristic optimization algorithm inspired by the natural behavior of elephant herds. It utilizes an information feedback model to adjust the distances of elephants within each clan in relation to their maternal elephant's position. The efficacy of the EHO method has been validated across various benchmark problems (*Li et al., 2020*), establishing its superiority in optimization. *Shao & Fan (2021)* introduced the BAS algorithm, which incorporates an elite selection mechanism and a neighbor mobility strategy. Initially, the algorithm computes the Euclidean distance between individual fitness values and the optimal individual fitness. If this distance falls below a predetermined threshold, individual positions are dynamically updated to enhance population diversity. Elite individuals, showing strong convergence and robustness, are then chosen to lead other individuals in exploring better positions. In a similar context, *Qin et al. (2023)* proposed the Historical Information-based Differential Evolution (HIDE) algorithm. This algorithm introduces a mechanism to identify individuals in a stagnant state and a mutation strategy based on discarded parent vectors from different time periods, enabling stagnant individuals to escape local optima. Moreover, a novel control parameter update method based on historical information was devised.

This article presents a new algorithm for identifying maximum $\gamma$-quasi-clique, inspired by the NuQClq algorithm and incorporating an information feedback model. While this model has been widely used in various fields, its application to guide $\gamma$-quasi-clique search algorithms is a unique approach. The subsequent sections of this study explore the secondary scoring function, information feedback model, and two newly proposed algorithms. Experimental analyses are performed on these variations to evaluate their effectiveness. The organization of the following sections is as follows: first, we discuss the secondary scoring function; second, we focus on the information feedback model. Then,

we introduce the NF1 and NR1 algorithms. The experimental results are detailed in the Experiment section. Finally, conclusions are drawn to summarize the key findings.

## SECONDARY SCORING FUNCTION

Before introducing the information feedback model, it is important to first explain the secondary scoring function in NuQClq. In the main scoring function of the previous MQCP heuristic algorithm, many vertices often have identical scores, leading to a higher risk of results converging to local optima. To address this issue, the author introduced a secondary scoring function based on the concept of cumulative saturation. This function calculates the secondary score of a vertex by considering variables such as $d_s(v)$ and recent trends, aiding in the decision to add or remove the vertex.

They are based on the number of endpoints included in $S$ (recorded as $\lambda(e)$) to distinguish each edge, *i.e.* for any edge, There are three possible values.

- $\lambda(e) = 0$ means the endpoint of $e$ is not included in $S$.
- $\lambda(e) = 1$ means that $S$ contains only one endpoint of $e$. if and only if $\lambda(e) = 1$, an edge is called *critical*.
- $\lambda(e) = 2$ means that both endpoints of $E$ are in $S$. if and only if $\lambda(e) = 2$, this edge is called *full*.

Encouraging critical edges to transform into *full* edges is essential for finding the maximum $\gamma$-quasi-clique. The main scoring function overlooks the results of intermediate steps, hindering the ability to obtain better results. To address this, the intermediate process should be recorded and analyzed. This is achieved by defining a vertex property known as saturation. In the local search algorithm, the saturation of vertex $v$ at step $t$ in the input graph $G = (V, E)$ is calculated as follows:

$$\Gamma_t(v) = \sum_{e \in E(v)} I(e) \tag{1}$$

$I(e)$ is an indicator of *critical*, i.e $\lambda(e) = 1$, $I(e) = 1$, otherwise $I(e) = 0$. Based on the concept of saturation, a cumulative saturation function is designed as the secondary scoring function of NuQClq algorithm:

$$\Gamma(v) = \sum_{t=t_0(v)}^{T} \Gamma_t(v) - T_S(v) \cdot \Delta G \tag{2}$$

where $t$ is the current number of steps, $t_0$ is the number of steps that vertex $v$ changed its state (added or deleted) last time, and $T_S(v)$ is the number of steps of vertex $v$ since $t_0$, $\Delta G$ is the maximum degree of vertex $v$ in the vertex set $V$. At the current step $t$, there are two situations.

**Situation 1** $v \notin S$

In this situation, $T_S(v) = 0$. So, $\Gamma(v)$ is the sum of $\Gamma_t(v)$ over steps since $t_0(v)$. And we prefer to add vertices with bigger $\Gamma(v)$.

**Situation 2** $v \in S$

In this situation, $\Gamma(v) \leq 0$. And we prefer to remove vertices with smaller $\Gamma(v)$ (*i.e.*, bigger $|\Gamma(v)|$).

Based on the above situation, the addition rules and deletion rules of vertices are obtained respectively.

### Add rule

Select the vertex with the highest $d_s(v)$ value ( $d_s(v)$ indicates the number of vertices connected to vertex $v$ in $S$), and select the vertex with the highest $\Gamma(v)$ value to break the ties, and further ties will be broken randomly.

### Delete rule

select the vertex with the lowest $d_s(v)$ value, and select the vertex with the highest $|\Gamma(v)|$ value to break the ties, and further ties will be broken randomly.

The concept of secondary scoring function is an important foundation for our newly proposed two variants based on the information feedback model. Next, we will introduce the information feedback model.

## INFORMATION FEEDBACK MODEL

The MQCP heuristic algorithm progresses through various stages: parameter and solution set initialization, iterative search, and result output. However, in the iterative search phase typical of many metaheuristic algorithms, the valuable insights gathered from vertices in previous iterations are frequently overlooked. Leveraging this information during the search process offers a promising opportunity for significantly enhancing the quality of the final results. *Wang & Tan (2017)* introduced an information feedback model aimed at leveraging insights from previous vertices to guide subsequent searches. Using a simple fitness weighting approach, individuals in the current iteration are updated with basic algorithms and selected past individuals. Through experimental validation, the authors demonstrated the superior performance of the variant over the basic algorithm on 14 standard test functions and 10 real-world problems from CEC2011. Several algorithms have been improved by integrating information feedback models, such as the MOEA/D algorithm. *Zhang et al. (2020)* further enhanced this with their MOEA/D-IFM algorithm, incorporating information feedback models and a new selection strategy for better performance. On the other hand, while NSGA-III excels in multi-objective optimization, it struggles with large-scale problems. To address this, *Gu & Wang (2020)* integrated an information feedback model into NSGA-III, significantly enhancing its capabilities for large-scale optimization. Experimental results confirm the competitive performance of their proposed algorithm across various test problems.

Assuming $x_i^t$ is the $i$-th individual in the $t$-th iteration, and $f_i^t$ is its fitness value. Here $t$ is the current iteration, $1 \leq i \leq K$ is an integer, $K$ is the population size, and in the MQCP problem, $K$ is the number of vertices. $y_i^{t+1}$ is the individual generated by the original algorithm, and $f_i^{t+1}$ is its fitness. The framework of the information feedback model is provided by individuals in the $t-2$, $t-1$, $t$ and $t+1$ iterations. By selecting different numbers of individuals, the model can be extended to different forms such as F1, R1, F2, R2, etc. Below are examples of F1 and R1 types.

**F1 and R1**

This is the simplest case. The $i$-th individual can be generated as follows:

$$x_i^{t+1} = \alpha y_i^{t+1} + \beta x_j^t \tag{3}$$

where $x_j^t$ is the position of individual $j(j \in \{1, 2, \ldots, K\})$ at $t$, and $f_j^t$ is its fitness. $\alpha$ and $\beta$ are weighting factors that satisfy $\alpha + \beta = 1$. They can be given as:

$$\alpha = \frac{f_j^t}{f_i^{t+1} + f_j^t}, \beta = \frac{f_i^{t+1}}{f_i^{t+1} + f_j^t} \tag{4}$$

Individual $j$ can be determined by the following definition.

**Definition 1**

When $j = i$, the model in Eq. (3) is called model F1.

**Definition 2**

When $j = m$, the model in Eq. (3) is called model R1, where $m$ is a randomly selected individual between 1 and $K$, excluding $i$.

The population diversity of individuals generated by Definition 2 is higher than that of individuals generated by Definition 1. When $m = i$, Model R1 will have an F1 probability of $1/K$. The binding of these individuals with the NuQClq algorithm results in NF1 and NR1, respectively. The information feedback model $q$ (where $q$ represents the number of selected individuals) theoretically has the ability to select previous individuals, but the method becomes more complex when a large number of individuals are used. Experimental results show that when $q \geq 2$, the outcomes are very unsatisfactory. Hence, this article showcases the improved algorithms NF1 and NF2 when $q = 1$. Our algorithm flowchart is shown in Fig. 1.

# NF1 AND NR1

The local search algorithm is a heuristic approach used to solve optimization problems. This method iteratively explores the solution space from a starting point to improve solution quality. Unlike exact algorithms that ensure optimality but may face challenges with large-scale instances, local search algorithms provide faster, approximate solutions, making them suitable for larger problems. In optimization, these algorithms are widely applied for navigating complex solution spaces and offering reasonable solutions within a reasonable time.

In many scenarios, we cannot get the global state of the graph. In this case, the artificial intelligence approach of local search can effectively solve the problem because it only needs local information to perform the search operation. When solving the maximum quasi-clique problem, the local search algorithm can adjust the quality of the solution by adding or removing vertices according to the current state, without the support of global graph information. The advantage of this local search method is that it can quickly find the local optimal solution without complete graph information, so it is practical and feasible in solving practical problems. A local search algorithm follows a basic process: starting with a

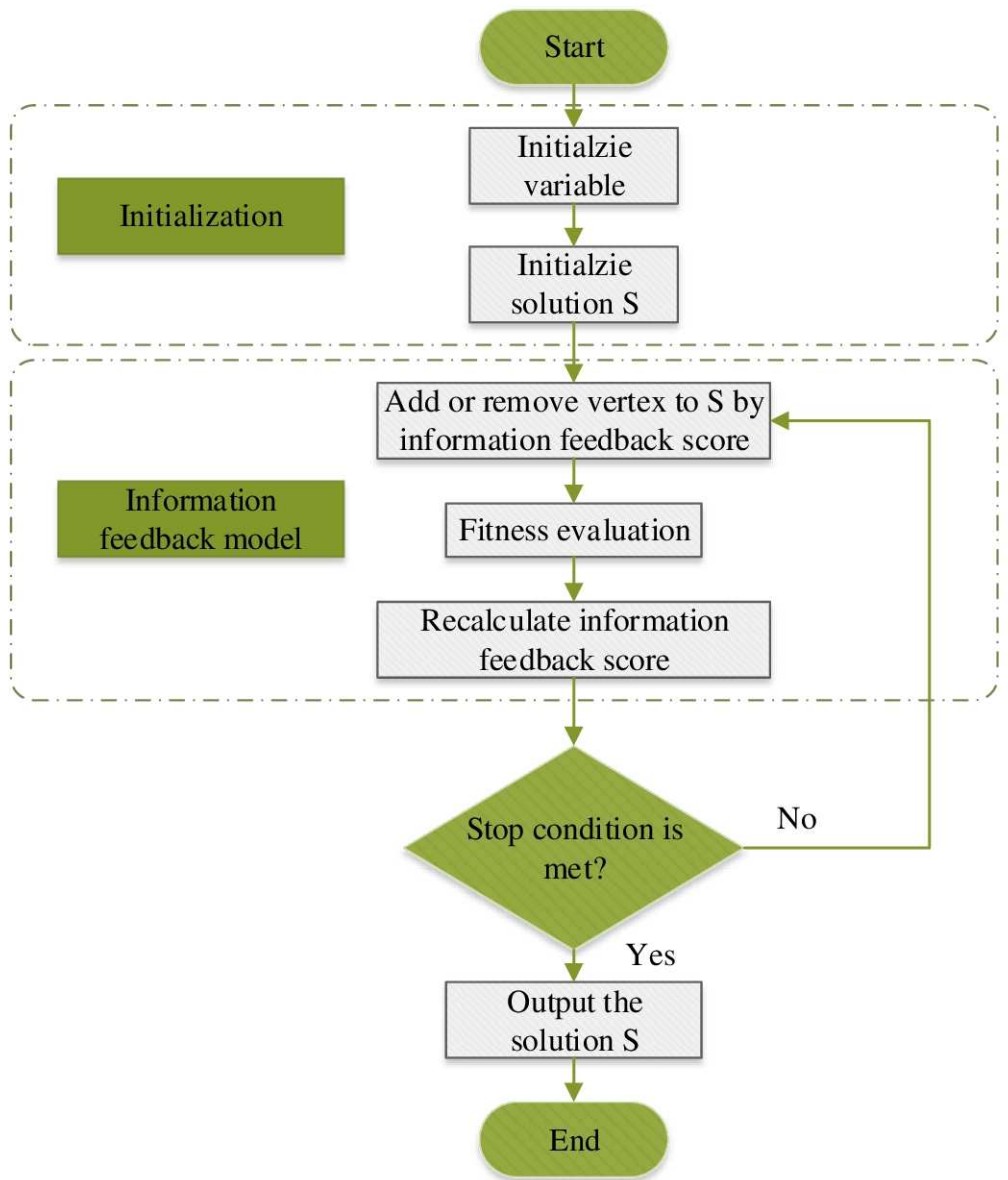

**Figure 1 Flowchart for two variants.**

randomly generated solution, it uses a specific strategy to gradually improve this solution within the solution space, with the goal of finding a better solution. After a set number of iterations, if no better solution is found, the algorithm either stops or starts a new search with a new randomly generated solution. The key element of the algorithm is the vertex selection mechanism, which determines the search direction and impacts the effectiveness of the overall local search strategy.

The information feedback model adopts specific calculation methods for different parameters in different problem domains. For example, the particle swarm optimization algorithm (*Wang & Tan, 2017*) incorporates inertial parameters in fitness calculation to

control flying dynamics. On the other hand, when addressing the maximum quasi-clique problem, vertex degree becomes a crucial parameter for evaluating fitness. However, relying solely on vertex degrees often leads to suboptimal results. The NuQClq algorithm proposed by *Chen et al. (2021)* introduces a comprehensive evaluation parameter that proves to be more effective. The algorithm's secondary scoring function captures important information related to vertices in the maximum quasi-clique problem, including neighboring points' degrees and vertex degrees. In our enhanced algorithm, we have implemented two two-dimensional arrays, namely *scoreHistory* and *populationHistory*, to meticulously record the secondary scores assigned to vertices by the NuQClq algorithm in each iteration, along with the quasi-clique size. By leveraging this compiled information, we carefully select the vertices to participate in subsequent iterations.

**Model 1:** Imformation Feedback F1
**Input:** a graph $G = (V, E)$; current iteration rounds; vertex $v$
**Output: imformation feedback score;**
1. while current iteration round $\geq 3$ do
2. calculate $f_i^t$ and $f_i^{t+1}$ with scoreHistory($v$);
3. calculate $\alpha$ and $\beta$ with $f_i^t$ and $f_i^{t+1}$;
4. calculate **imformation feedback score** with $\alpha$, $\beta$ and populationHistory($v$);
5. return **imformation feedback score**.

**Model 2:** Imformation Feedback R1
**Input:** a graph $G = (V, E)$; current iteration rounds; vertex $v$
**Output: imformation feedback score;**
1. while current iteration round $\geq 3$ do
2. select $u \in V$ randomly;
3. calculate $f_j^t$ and $f_i^{t+1}$ with scoreHistory($v$);
4. calculate $\alpha$ and $\beta$ with $f_j^t$ and $f_i^{t+1}$;
5. calculate **imformation feedback score** with $\alpha$, $\beta$, populationHistory ($u$) and populationHistory($v$);
6. return **imformation feedback score**.

The calculation of F1 and F2 begins when the current iteration count is 3 or more. F1 computes $f_i^t$ and $f_i^{t+1}$ using *scoreHistory* values of vertex $v$ from the previous and current iterations. R1 randomly selects a vertex $u$ and computes $f_j^t$ and $f_i^{t+1}$ based on *scoreHistory* values of $u$ in the previous iteration and $v$ in the current iteration. The resulting fitness values are then used to derive $\alpha$ and $\beta$, which, along with *populationHistory* values from previous and current iterations, inform information feedback scores for guiding vertex selection or deletion strategies in the next iteration. Two variants of the NuQClq algorithm, NF1 and NR1, are derived from these models, as depicted in Algorithm 1 .

Utilizing the model outlined above, we delineate a comprehensive process for two variants of the NuQClq algorithm designed to tackle the maximum $\gamma$-quasi-clique problem: NF1 and NR1 (outlined in Algorithm 1). The algorithm initiates by establishing the selected vertex set, denoted as $S$ (line 1), employing the construction method from NuQClq to formulate viable solutions. Subsequently, a local search strategy is employed to refine these solutions. When S represents a $\gamma$-quasi-clique with an edge density not falling below $\gamma$, the algorithm endeavors to augment $S$ by adding more vertices. Each vertex, during addition or removal operations, maintains its secondary score for the current iteration scoreHistory($v$) and the size of the $\gamma$-quasi-clique populationHistory($v$). The algorithm then calculates the information feedback score, incorporating the F1 or R1 information feedback model. The vertex addition operation involves selecting the vertex with the highest score (line 4). Conversely, if $S$ does not constitute a $\gamma$-quasi-clique, the algorithm adds a vertex to $S$ by choosing the one with the highest information feedback score (line 9), while removal involves selecting the one with the lowest information feedback score (line 11). In cases where a suitable solution cannot be found within certain iterations and the specified execution time has not been reached, the algorithm initiates a restart (Line 14).

---

**Algorithm 1:** NF1 and NR1

---

**Input:** a graph $G = (V, E)$; the parameter $\gamma$

**Output:** the best $\gamma$-quasi-clique $S_{best}$;

1. $S \leftarrow$ an initial solution; current iteration round $\leftarrow 0$;

2. **while** current iteration round $< MaxRound$ **do**

3.     **if** $dens(G[S]) \geq \gamma$ **then**

4.         select a vertex $v$ by choosing highest **imformation feedback score,** and $S \leftarrow S \cup v$;

5.         update scoreHistory ($v$) and populationHistory($v$);

6.         update $S_{best}$ if a new better solution is found;

7.         current iteration round $\leftarrow 0$;

8. **else**

9.         select $v$ by choosing highest **imformation feedback score**; $S \leftarrow S \cup v$;

10.        update scoreHistory($v$) and populationHistory($v$);

11.        select $u$ by choosing lowest **imformation feedback score**; and $S \leftarrow S \setminus u$;

12.        update scoreHistory($u$) and populationHistory($u$);

13.        current iteration round $\leftarrow$ current iteration round $+1$;

14. **if** not reach the limited time **then** restart the algorithm;

15. **return** $S_{best}$.

---

In algorithms, the space complexity of the two-dimensional arrays *scoreHistory* and *populationHistory* depends on their number of rows and columns, which correspond to the maximum number of iterations and total number of vertices, respectively. Therefore, the spatial complexity is $O(M \cdot V)$. Vertex selection is mainly achieved through information feedback scoring, which involves tasks like comparing target values and updating result arrays. The time complexity of these operations is mainly influenced by the size of the

candidate array, leading to an overall algorithm time complexity of $O(M \cdot V)$, where $M$ is the maximum number of iterations and $V$ is the total number of vertices.

The algorithm records historical second-level rating information and maximum $\gamma$-quasi-clique information obtained from each iteration of every vertex. When the iteration number is 3 or higher, the information feedback rating is calculated, and vertex selection is based on this rating. If the number of iterations is less than 3, a vertex $v$ is randomly selected from the set to add to the candidate set, and its second-level score is calculated, along with recording the maximum cluster it forms. When the iteration reaches *MaxRound*, the process resets to 0 and continues until the restricted search time is reached.

## EXPERIMENT

Our novel variants and the NuQClq algorithm differ in their approach to enhancing search diversity. While NuQClq uses a secondary scoring function, it does not fully utilize information from previous vertices to guide the search comprehensively. On the other hand, our proposed variants consider the maximum $\gamma$-quasi-clique formed by historical vertices as crucial information. They incorporate the secondary scoring function into fitness evaluation, introducing a weighted calculation that includes the maximum clique formed by vertices from both current and previous iterations. This process generates an information feedback score, guiding the $\gamma$-quasi-clique search. In the following sections, we will present the results of experimental analyses conducted on the two variants and NuQClq.

Table 1 presents the experimental outcomes of NuQClq, NF1, and NR1 across 50 instances in dense instance. The search time was limited to a maximum of 1000 seconds per instance. NF1 demonstrated superior or comparable results in 48 instances, highlighting its effectiveness. However, exceptions were noted in the DIMACS benchmark graph C2000.9 and BIHOSLIB instance graph frb59-26-2, both with a density threshold $\gamma$ of 0.95, where NF1 slightly underperformed NuQClq. On the other hand, NR1, with a density threshold $\gamma$ of 0.95, showed a lower result for C2000.9 compared to NuQClq but performed better in the remaining 49 instances. Interestingly, both variants consistently achieved better or comparable results for all instances at a density threshold $\gamma$ of 0.999. These experimental results highlight the effectiveness of NF1 and NR1 in addressing the maximum $\gamma$-quasi-clique problem, particularly showing improvements at a higher density threshold of $\gamma = 0.999$. These advancements suggest the potential impact of these algorithms on real-world instances.

In light of the properties of sparse graphs, we opted to evaluate the efficiency of three algorithms at reduced quasi-clique density thresholds. Table 2 displays the outcomes of the experiments conducted on sparse graphs using these algorithms. The results suggest that NF1 and NF2 are capable of identifying the largest quasi-clique in sparse graphs. Additionally, we performed a statistical analysis on the degree distribution of vertices in both dense and sparse graphs, selecting 5 representative graphs of each type to capture the overall patterns. Figures 2 and 3 depict the vertex degree distribution of the sparsely and partially densely populated graphs that were examined.

**Table 1  Comparison of experimental results in dense instance.**

| Instance | $|V|$ | $|E|$ | $\gamma$ | NuQClq | NF1 | NR1 |
|---|---|---|---|---|---|---|
| Brock400_2 | 400 | 59786 | 0.999 | **29** | **29** | **29** |
| | | | 0.95 | **40** | **40** | **40** |
| Brock400_3 | 400 | 59681 | 0.999 | **31** | **31** | **31** |
| | | | 0.95 | **39** | **39** | **39** |
| C1000.9 | 1000 | 450079 | 0.999 | 69.9 | **70** | **70** |
| | | | 0.95 | **222** | 222 | 222 |
| C2000.9 | 2000 | 1799532 | 0.999 | 79.3 | **80.4** | 79.9 |
| | | | 0.95 | **288** | 285.5 | 285.4 |
| gen200_p0.9_44 | 200 | 17910 | 0.999 | **44** | **44** | **44** |
| | | | 0.95 | **104** | **104** | **104** |
| gen400_p0.9_55 | 400 | 71820 | 0.999 | **55** | **55** | **55** |
| | | | 0.95 | **183** | **183** | **183** |
| gen400_p0.9_65 | 400 | 71820 | 0.999 | **66** | **66** | **66** |
| | | | 0.95 | **197** | **197** | **197** |
| hamming10-4 | 1024 | 434176 | 0.999 | **40** | **40** | **40** |
| | | | 0.95 | 87.2 | **87.7** | **87.8** |
| frb50-23-1 | 1150 | 580603 | 0.999 | 50 | **50.3** | 50 |
| | | | 0.95 | **164** | **164** | **164** |
| frb50-23-2 | 1150 | 579824 | 0.999 | 50 | **50.2** | 50 |
| | | | 0.95 | **161** | **161** | **161** |
| frb50-23-4 | 1150 | 580417 | 0.999 | 50.9 | **51** | **51** |
| | | | 0.95 | **162** | **162** | **162** |
| frb50-23-5 | 1150 | 580640 | 0.999 | 50.7 | **50.9** | 50.7 |
| | | | 0.95 | **165** | **165** | **165** |
| frb53-24-1 | 1272 | 714129 | 0.999 | 52.8 | **52.9** | 52.8 |
| | | | 0.95 | **204** | **204** | **204** |
| frb53-24-2 | 1272 | 714067 | 0.999 | **53** | **53** | **53** |
| | | | 0.95 | **177** | **177** | **177** |
| frb53-24-4 | 1272 | 714048 | 0.999 | 52.3 | 52.7 | **52.8** |
| | | | 0.95 | **185** | **185** | **185** |
| frb53-24-5 | 1272 | 714130 | 0.999 | 52.3 | 52.9 | **53.1** |
| | | | 0.95 | **175** | **175** | **175** |
| frb59-26-1 | 1534 | 1049256 | 0.999 | 58.1 | **58.3** | 58.2 |
| | | | 0.95 | **248** | **248** | **248** |
| frb59-26-2 | 1534 | 1049648 | 0.999 | 58.1 | **58.3** | 58.2 |
| | | | 0.95 | **244** | 243.7 | 243.7 |
| frb59-26-4 | 1534 | 1048800 | 0.999 | 58 | **58.2** | 58.1 |
| | | | 0.95 | **238** | **238** | **238** |
| frb59-26-5 | 1534 | 1049829 | 0.999 | **58.6** | **58.6** | **58.6** |
| | | | 0.95 | **231** | **231** | **231** |
| p-hat1000-1 | 1000 | 122253 | 0.999 | **10** | **10** | **10** |
| | | | 0.95 | **12** | **12** | **12** |

**Table 1** (*continued*)

| Instance | $|V|$ | $|E|$ | $\gamma$ | NuQClq | NF1 | NR1 |
|---|---|---|---|---|---|---|
| p-hat1000-2 | 1000 | 244799 | 0.999 | 47 | 47 | 47 |
|  |  |  | 0.95 | 109 | 109 | 109 |
| p-hat1000-3 | 1000 | 371746 | 0.999 | 70 | 70 | 70 |
|  |  |  | 0.95 | 210 | 210 | 210 |
| p-hat1500-1 | 1500 | 284923 | 0.999 | 12 | 12 | 12 |
|  |  |  | 0.95 | 14 | 14 | 14 |
| p-hat1500-2 | 1500 | 568960 | 0.999 | 67 | 67 | 67 |
|  |  |  | 0.95 | 193 | 193 | 193 |

In the dense graph tested, there are no distinct discontinuities in the degree distribution of vertices. The degrees of vertices cover a range from 1 to the total number of vertices in each graph, with the highest degree almost reaching the total number of vertices. The NF1 and NR1 algorithms rely on historical vertex information for selection, and the dense graph ensures rich connection relationships between vertices, providing comprehensive vertex information. This allows NF1 and NR1 to effectively leverage this rich connectivity for better results. In contrast, sparse graphs exhibit discontinuous degree distributions and weaker vertex connections, resulting in less significant impact for NF1 and NR1, although results comparable to NuQClq can still be achieved.

Our proposed variants utilize two variables, *scoreHistory* and *populationHistory*, to track the historical secondary scores and the maximum historical $\gamma$-quasi-clique of each vertex across different iterations. The number of maximum iterations in the algorithm plays a crucial role in influencing the results. Figures 4 and 5 depict the impact of variant F1 and variant R1 on the algorithm at varying maximum iterations, respectively.

We have analyzed various examples using different maximum iterations such as 4,000, 2,000, and 1,000. Our findings show a noticeable variance in results across these iterations. Specifically, when comparing F1 and R1 scores, the 4,000 iteration mark outperforms NuQClq. Consequently, we have determined that setting the maximum number of iterations to 4,000 in the algorithm yields superior outcomes.

Table 3 illustrates the search performance of each algorithm across different instances. Despite my algorithm displaying a slightly higher average search time compared to the comparison algorithms, all results remain well below the maximum search time limit of 1000 seconds. This indicates that while there may be a slight increase in search time in certain scenarios, overall it remains within an acceptable range and does not exceed the predetermined time limit, thereby not significantly impacting the algorithm's practicality. Moving forward, reducing search time will be a key focus for our future research endeavors. In general, the discovery of larger $\gamma$-quasi-cliques holds vast potential for the analysis and optimization of diverse complex systems. This contributes to a deeper understanding of the internal structure and relationships within the system, offering more effective methods and strategies for addressing real-world problems.

**Table 2 Comparison of experimental results in sparse instance.**

| Instance | $|V|$ | $|E|$ | $\gamma$ | NuQClq | NF1 | NR1 |
|---|---|---|---|---|---|---|
| bio-WormNet | 2444 | 78731 | 0.9 | 136 | 136 | 136 |
| | | | 0.8 | 150 | 150 | 150 |
| | | | 0.7 | 176 | 176 | 176 |
| | | | 0.6 | 199 | 199 | 199 |
| | | | 0.5 | 241 | 241 | 241 |
| CSphd | 1882 | 1740 | 0.8 | 4 | 4 | 4 |
| | | | 0.7 | 5 | 5 | 5 |
| | | | 0.6 | 6 | 6 | 6 |
| delaunay_n10 | 1024 | 3056 | 0.7 | 6 | 6 | 6 |
| | | | 0.6 | 7 | 7 | 7 |
| | | | 0.5 | 9 | 9 | 9 |
| EX2 | 560 | 4368 | 0.6 | 6 | 6 | 6 |
| | | | 0.5 | 9 | 9 | 9 |
| G44 | 1000 | 9990 | 0.7 | 5 | 5 | 5 |
| | | | 0.6 | 6 | 6 | 6 |
| 145bit | 1002 | 11315 | 0.9 | 8 | 8 | 8 |
| | | | 0.8 | 10 | 10 | 10 |
| | | | 0.7 | 13 | 13 | 13 |
| | | | 0.6 | 17 | 17 | 17 |
| | | | 0.5 | 22 | 22 | 22 |
| ia-email-univ | 1133 | 5451 | 0.9 | 13 | 13 | 13 |
| | | | 0.8 | 15 | 15 | 15 |
| | | | 0.7 | 17 | 17 | 17 |
| | | | 0.6 | 20 | 20 | 20 |
| | | | 0.5 | 25 | 25 | 25 |
| ia-fb-message | 1266 | 6451 | 0.7 | 9 | 9 | 9 |
| | | | 0.6 | 11 | 11 | 11 |
| | | | 0.5 | 16 | 16 | 16 |
| inf-euroroad | 1171 | 1417 | 0.7 | 4 | 4 | 4 |
| | | | 0.6 | 5 | 5 | 5 |
| | | | 0.5 | 6 | 6 | 6 |
| soc-hamsterster | 2426 | 16630 | 0.9 | 29 | 29 | 29 |
| | | | 0.8 | 32 | 32 | 32 |
| | | | 0.7 | 40 | 40 | 40 |
| | | | 0.6 | 49 | 49 | 49 |
| | | | 0.5 | 63 | 63 | 63 |
| soc-FourSquare | 639014 | 3214986 | 0.9 | 58 | 58 | 58 |
| | | | 0.8 | 76 | 76 | 76 |
| | | | 0.7 | 95 | 95 | 95 |
| | | | 0.6 | 117 | 117 | 117 |

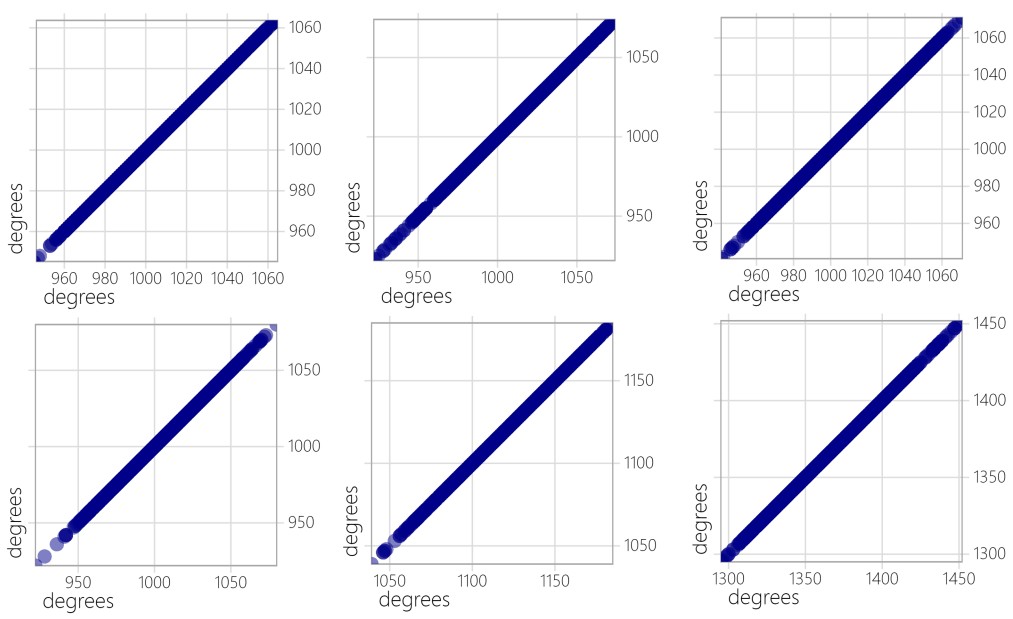

**Figure 2** Degree distribution of dense graphs.

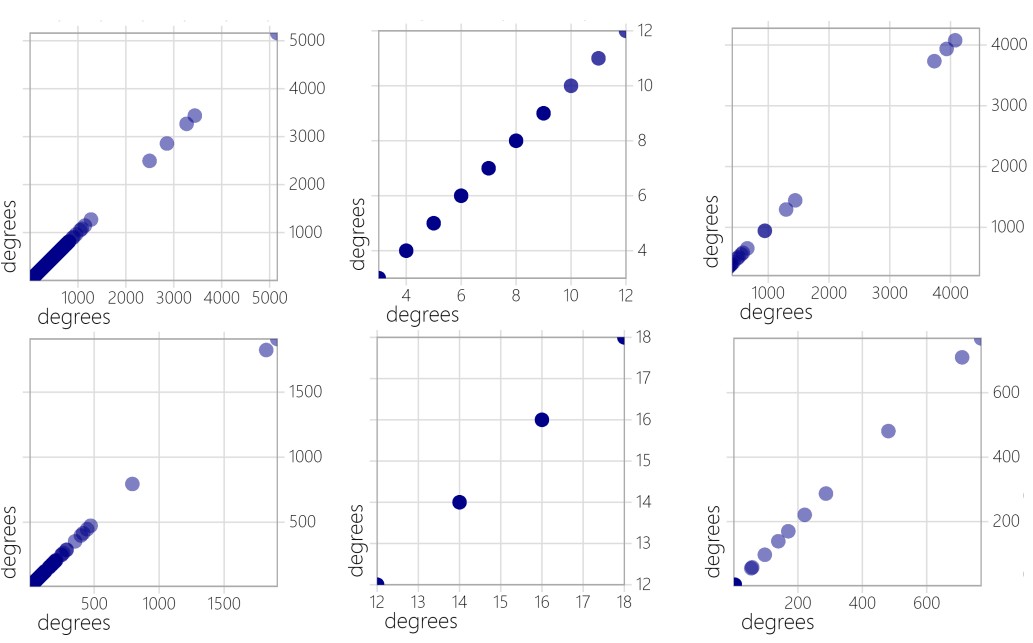

**Figure 3** Degree distribution of sparse graphs.

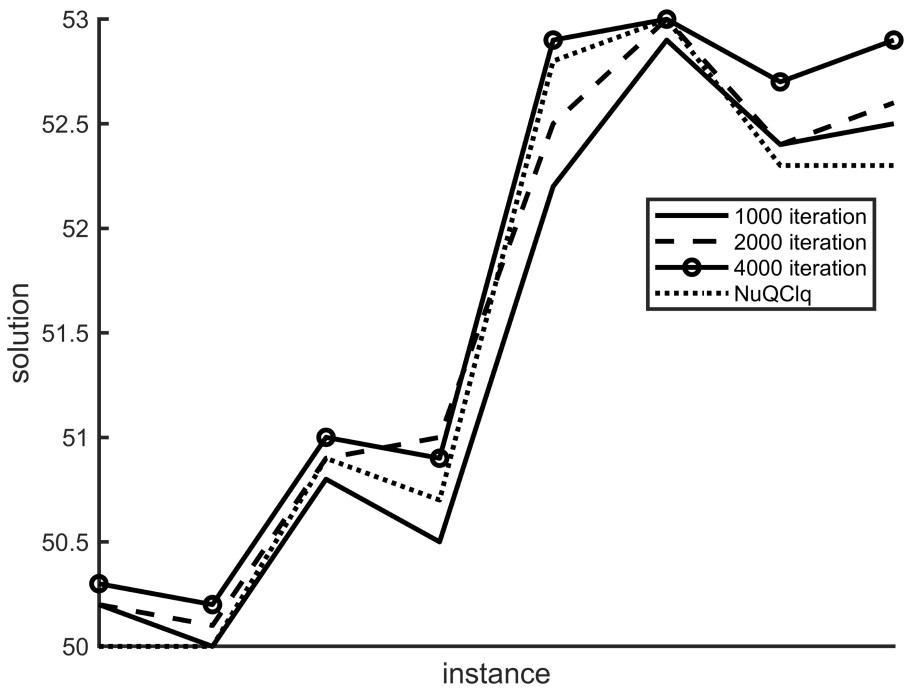

**Figure 4** The results of variant NF1 under different iterations.

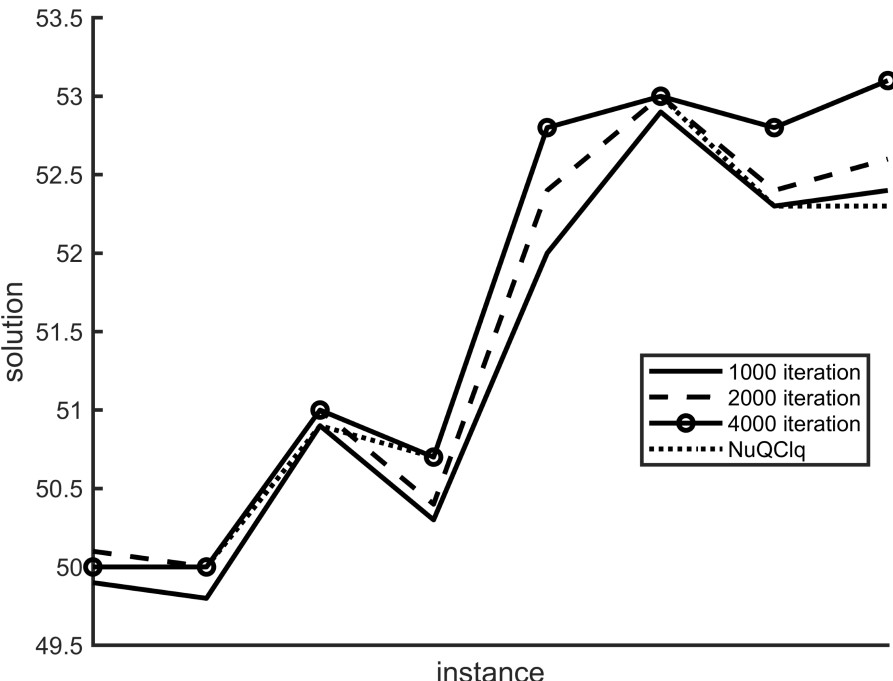

**Figure 5** The results of variant NR1 under different iterations.

**Table 3  The search time for each instance.**

| Instance | $\gamma$ | NuQClq | NF1 | NR1 | Instance | $\gamma$ | NuQClq | NF1 | NR1 |
|---|---|---|---|---|---|---|---|---|---|
| Brock400_2 | 0.999 | 18.05 | 46.65 | 41.13 | | 0.9 | 0.01 | 0.19 | 0.24 |
| | 0.95 | 0.01 | 0.17 | 0.24 | | 0.8 | 0.13 | 0.61 | 0.46 |
| Brock400_3 | 0.999 | 37.24 | 61.19 | 103.18 | bio-WormNet | 0.7 | 0.11 | 0.61 | 0.33 |
| | 0.95 | 0.01 | 0.09 | 0.03 | | 0.6 | 0.07 | 0.39 | 0.14 |
| C1000.9 | 0.999 | 202.84 | 54.87 | 27.23 | | 0.5 | 187.98 | 235.05 | 219.00 |
| | 0.95 | 0.66 | 34.05 | 42.42 | | 0.8 | 0.01 | 0.01 | 0.01 |
| C2000.9 | 0.999 | 290.28 | 450.15 | 292.39 | CSphd | 0.7 | 0.01 | 0.01 | 0.01 |
| | 0.95 | 49.32 | 200.76 | 282.06 | | 0.6 | 0.01 | 0.01 | 0.01 |
| gen200_p0.9_44 | 0.999 | 0.01 | 0.03 | 0.01 | | 0.7 | 0.01 | 0.01 | 0.01 |
| | 0.95 | 0.01 | 0.01 | 0.01 | delaunay_n10 | 0.6 | 0.01 | 0.01 | 0.01 |
| gen400_p0.9_55 | 0.999 | 0.11 | 0.49 | 3.41 | | 0.5 | 0.01 | 0.01 | 0.01 |
| | 0.95 | 0.01 | 0.03 | 0.04 | EX2 | 0.6 | 0.01 | 0.01 | 0.01 |
| gen400_p0.9_65 | 0.999 | 0.01 | 0.05 | 0.05 | | 0.5 | 0.01 | 0.01 | 0.01 |
| | 0.95 | 0.02 | 0.15 | 0.12 | G44 | 0.7 | 0.01 | 0.01 | 0.01 |
| hamming10-4 | 0.999 | 0.05 | 0.15 | 0.08 | | 0.6 | 0.01 | 0.01 | 0.01 |
| | 0.95 | 243.62 | 186.65 | 168.25 | | 0.9 | 0.01 | 0.02 | 0.01 |
| frb50-23-1 | 0.999 | 203.74 | 158.31 | 140.48 | | 0.8 | 0.01 | 0.01 | 0.01 |
| | 0.95 | 1.85 | 45.55 | 19.19 | 145bit | 0.7 | 0.01 | 0.01 | 0.02 |
| frb50-23-2 | 0.999 | 270.72 | 130.18 | 56.04 | | 0.6 | 0.02 | 0.03 | 0.03 |
| | 0.95 | 1.13 | 10.53 | 25.66 | | 0.5 | 0.01 | 0.01 | 0.01 |
| frb50-23-4 | 0.999 | 223.51 | 179.90 | 148.98 | | 0.9 | 0.01 | 0.03 | 0.01 |
| | 0.95 | 1.41 | 41.64 | 69.14 | | 0.8 | 0.01 | 0.07 | 0.08 |
| frb50-23-5 | 0.999 | 258.83 | 326.16 | 447.68 | ia-email-univ | 0.7 | 0.03 | 0.06 | 0.05 |
| | 0.95 | 0.69 | 35.19 | 59.85 | | 0.6 | 0.02 | 0.01 | 0.03 |
| frb53-24-1 | 0.999 | 279.06 | 317.25 | 100.68 | | 0.5 | 0.01 | 0.01 | 0.01 |
| | 0.95 | 1.72 | 40.52 | 29.76 | | 0.7 | 0.01 | 0.01 | 0.01 |
| frb53-24-2 | 0.999 | 266.89 | 240.48 | 140.37 | ia-fb-message | 0.6 | 0.01 | 0.01 | 0.01 |
| | 0.95 | 0.39 | 15.59 | 18.40 | | 0.5 | 0.01 | 0.01 | 0.01 |
| frb53-24-4 | 0.999 | 97.20 | 233.13 | 404.01 | | 0.7 | 0.01 | 0.01 | 0.01 |
| | 0.95 | 1.38 | 42.42 | 46.22 | inf-euroroad | 0.6 | 0.01 | 0.01 | 0.01 |
| frb53-24-5 | 0.999 | 116.62 | 204.09 | 331.65 | | 0.5 | 0.01 | 0.01 | 0.01 |
| | 0.95 | 1.78 | 108.43 | 138.65 | | 0.9 | 0.12 | 0.55 | 0.34 |
| frb59-26-1 | 0.999 | 127.64 | 172.96 | 179.37 | | 0.8 | 0.01 | 0.01 | 0.01 |
| | 0.95 | 6.07 | 46.78 | 145.21 | soc-hamsterster | 0.7 | 0.01 | 0.01 | 0.01 |
| frb59-26-2 | 0.999 | 39.28 | 153.32 | 60.76 | | 0.6 | 0.01 | 0.01 | 0.01 |
| | 0.95 | 12.25 | 222.48 | 207.89 | | 0.5 | 0.01 | 0.01 | 0.01 |
| frb59-26-4 | 0.999 | 159.88 | 148.15 | 116.94 | | 0.9 | 3.94 | 25.09 | 77.87 |
| | 0.95 | 2.67 | 209.62 | 198.92 | soc-FourSquare | 0.8 | 7.77 | 30.24 | 64.26 |
| frb59-26-5 | 0.999 | 268.93 | 189.39 | 182.27 | | 0.7 | 10.24 | 24.26 | 51.30 |
| | 0.95 | 1.57 | 34.07 | 65.09 | | 0.6 | 13.55 | 16.17 | 35.55 |

**Table 3** (*continued*)

| Instance | $\gamma$ | NuQClq | NF1 | NR1 | Instance | $\gamma$ | NuQClq | NF1 | NR1 |
|---|---|---|---|---|---|---|---|---|---|
| p-hat1000-1 | 0.999 | 0.01 | 0.05 | 0.01 | American75 | 0.999 | 0.69 | 1.41 | 2.79 |
| | 0.95 | 0.01 | 0.01 | 0.01 | | 0.95 | 0.98 | 1.30 | 1.65 |
| p-hat1000-2 | 0.999 | 0.01 | 0.01 | 0.01 | Auburn71 | 0.999 | 3.16 | 7.98 | 12.22 |
| | 0.95 | 0.01 | 0.01 | 0.01 | | 0.95 | 3.41 | 5.03 | 5.84 |
| p-hat1000-3 | 0.999 | 0.02 | 0.13 | 0.26 | Baylor93 | 0.999 | 1.32 | 2.33 | 4.08 |
| | 0.95 | 0.01 | 0.01 | 0.01 | | 0.95 | 1.06 | 2.59 | 2.00 |
| p-hat1500-1 | 0.999 | 7.50 | 22.65 | 18.46 | Berkeley13 | 0.999 | 2.64 | 6.50 | 7.96 |
| | 0.95 | 0.36 | 0.12 | 0.24 | | 0.95 | 11.63 | 19.65 | 14.00 |
| p-hat1500-2 | 0.999 | 0.01 | 0.03 | 0.08 | Brandeis99 | 0.999 | 0.48 | 0.99 | 1.23 |
| | 0.95 | 0.01 | 0.01 | 0.01 | | 0.95 | 0.92 | 1.72 | 1.58 |

# CONCLUSIONS AND FUTURE RESEARCH

This study explores the maximum $\gamma$-quasi-clique problem and presents two novel local search algorithms, NF1 and NR1. These algorithms leverage an information feedback model that incorporates historical data from the NuQClq algorithm to calculate fitness values. Additionally, we consider the historical maximum $\gamma$-quasi-clique as important information for calculating feedback ratings. By strategically adding and deleting vertices, these algorithms offer effective solutions for addressing the maximum $\gamma$-quasi-clique problem using local search methods. The performance of NF1 and NR1 is evaluated through computational experiments on synthetic graphs and real-world network graphs across various applications, illustrating their efficacy.

We conducted a series of tests on very dense instances and sparse graphs, and obtained better experimental results in dense graphs. Although larger $\gamma$-quasi-clique cannot be obtained in sparse graphs, and this algorithm is only more effective in very dense situations, there are certain limitations to this approach. However, it cannot be denied that incorporating information feedback models into the maximum $\gamma$-quasi-clique search algorithm is a good optimization approach, achieving better or comparable results in almost all instances tested. For tests under other density thresholds and more instances, and further optimization of the algorithm will be important in our future work.

A series of tests were conducted on both very dense instances and sparse graphs, with better experimental results observed in dense graphs. It was noted that larger $\gamma$-quasi-cliques could not be obtained in sparse graphs, limiting the effectiveness of this method in such scenarios. However, it is undeniable that incorporating information feedback model into the maximum quasi-clique search algorithm is a feasible optimization method, has led to improved or equivalent maximum quasi-clique results in almost all tested instances. Further optimization of the algorithm will be a primary focus of our future research.

### Funding

This work was supported by the National Natural Science Foundation of China (No. 61862051 and No. 62241206), the Science and Technology Plan Project of Guizhou Province (No.(ZK[2022]449 and No. ZK[2022]550), the Natural Science Foundation of Education of Guizhou province (No. [2019]203) and the program of Qiannan Normal University for Nationalities (No. 2024zdzk03). There was no additional external funding received for this study. The funders had no role in study design, data collection and analysis, decision to publish, or preparation of the manuscript.

### Grant Disclosures

The following grant information was disclosed by the authors:
The National Natural Science Foundation of China: 61862051, 62241206.
The Science and Technology Plan Project of Guizhou Province: ZK[2022]449, ZK[2022]550.
The Natural Science Foundation of Education of Guizhou province: [2019]203.
The program of Qiannan Normal University for Nationalities: 2024zdzk03.

### Competing Interests

The authors declare there are no competing interests.

### Author Contributions

- Shuhong Liu conceived and designed the experiments, performed the experiments, analyzed the data, performed the computation work, prepared figures and/or tables, authored or reviewed drafts of the article, and approved the final draft.
- Jincheng Zhou conceived and designed the experiments, performed the experiments, analyzed the data, performed the computation work, prepared figures and/or tables, authored or reviewed drafts of the article, and approved the final draft.
- Dan Wang performed the experiments, analyzed the data, authored or reviewed drafts of the article, and approved the final draft.
- Zaijun Zhang performed the experiments, analyzed the data, authored or reviewed drafts of the article, and approved the final draft.
- Mingjie Lei performed the experiments, analyzed the data, prepared figures and/or tables, and approved the final draft.

### Data Availability

   Data, algorithm code, and computer code are available in the Supplemental Files.

### Supplemental Information

Supplemental information for this article can be found online at http://dx.doi.org/10.7717/peerj-cs.2173#supplemental-information.

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
