# Peer review of "An optimization algorithm for maximum quasi-clique problem based on information feedback model"

_PeerJ Computer Science, doi:10.7717/peerj-cs.2173_

## Round 0.1 · original submission · Major Revisions

Both reviewers raised critical comments. As a result, the current version cannot be accepted.

**Language Note:** The review process has identified that the English language must be improved. PeerJ can provide language editing services - please contact us at [email protected] for pricing (be sure to provide your manuscript number and title). Alternatively, you should make your own arrangements to improve the language quality and provide details in your response letter. – PeerJ Staff

Reviewer 1 ·

Basic reporting

In this paper, the authors propose the NuQClqF1 and NuQClqR1 algorithms, and applies them to guide γ- quasi - clique search algorithm. Compared with the existing NuQClq algorithm, the proposed algorithm adds a simple fitness weighting method to update individuals in the current iteration. Experimental studies have already demonstrated the superiority of the proposed strategy and algorithm. Although the proposed NuQClqF1 andNuQclqR1algorithm seems meaningful, it still lacks sufficient theoretical and experimental support. Furthermore, the proposed model is only the simple combination of existing model. In addition, the expression of this manuscript is fair and wordy. The language should be greatly improved.

Experimental design

More comparative experiments with benchmarks need to be conducted to further illustrate the advantages of this method.

Validity of the findings

This study only conducted a series of tests on γ-quasi-clique with strong constraints, and how to prove the effectiveness of the proposed method needs further clarification.

Additional comments

Major Points
(1) According to the experimental data in Table 1, the two methods proposed in this paper are not significantly superior to the NuQClq algorithm, and it can be seen from Table 2 that the proposed methods are more time-consuming.

(2) This study only conducted a series of tests on γ-quasi-clique with strong constraints, and how to prove the effectiveness of the proposed method needs further clarification.

(3) In the experimental part, more comparative experiments with benchmarks need to be conducted to further illustrate the advantages of this method.

Minor Points
(1) The expression of Abstract is fair and wordy. The language should be greatly improved.

(2) It is recommended to use serial number to mark the organizational structure of the article clearly.

(3) Some formats need to be standardized. Such as in Line 65 "(G[S]) ≥γ"; In Line 268 "NP"; And in Line 271 "in the (t_2), (t_1), t and(t+1) iterations". Double-check the whole manuscript.

(4) In Table 1, it is recommended to use the standard three-wire Table.

·

Basic reporting

68-> Related instead of Relate
325 -> Information instead of Imformation
326 -> Same as above

Experimental design

The benchmarks are incorrectly run. The average time is of less interest than the average time over many evaluations. Perhaps ASV (airspeed velocity) could be used. At the very least a graph of the time taken per problem per algorithm with agglomeration is required.

Validity of the findings

In some sense, new information of course is always useful to optimization algorithms. However, the costs and tradeoffs need to better explained. There is only a quick mention in 385 about the search time. Please quantify this cost more concretely, how much history is required? What is the complexity (time / memory)? Also, for a max iteration based approach, I would request that you consider the case of low and high max iterations.

---

## Round 0.2 · Major Revisions

The reviewer sees a major improvement in the manuscript. Yet, there are still several shortcomings that led me to a major revision. In particular, please address:
- Introduction of important AI techniques used
- Include a discussion of more recent literature related to your topic.

Reviewer 1 ·

Basic reporting

The proposed scheme is presented in great detail, but the language needs substantial improvement.

Experimental design

The experimental design is logical and interesting.

Validity of the findings

The findings are neutral, and if possible, more state-of- the-art models should be introduced for comparison.

Additional comments

1). The format of the manuscript needs to be further modified, and there are still many issues, such as: the first sentence of each paragraph needs to be indent, the line spacing of the whole paper is not uniform, such as “lines 59-65,375-378”, and the format of Algorithm1 is not aligned, such as lines 9-10.

2) The organizational structure of this paper is somewhat chaotic, and the authors need to reconsider. For example, the contents of "1. introduction" and "1.1 Relate Work" are usually placed in two different sections.

3). The cited references are not advanced enough, especially the number of articles cited in recent years is seriously insufficient, which cannot fully support the innovation of the method proposed in this study.

4) The "Figure 1" suggestion can be changed to a pictograph, which will be more intuitive; "Figure 2" and "Figure 3" need to improve the graphics quality.

---

## Round 0.3 · accepted · Accept

Thank you! I believe you addressed all comments from the AE and the reviewers adequately. The manuscript is ready for publication.